# Biomass fuel use and birth weight among term births in Nigeria

**Musa Abubakar Kana** [1,2,3,4]*, **Min Shi**[1], **Jennifer Ahmed**[4], **Jimoh Muhammad Ibrahim**[3,4], **Abdullahi Yusuf Ashir**[4], **Karimatu Abdullahi**[5], **Halima Bello-Manga**[3,4], **Matthew Taingson**[3,4], **Amina Mohammed-Durosinlorun**[3,4], **Musa Shuaibu**[3,4], **Abdulkadir Musa Tabari**[3,4], **Stephanie J. London** [1]

**1** National Institute of Environmental Health Sciences, National Institutes of Health, Department of Health and Human Services, Research Triangle Park, Durham, North Carolina, United States of America, **2** Department of Epidemiology and Community Medicine, Federal University of Lafia, Lafia, Nasarawa State, Nigeria, **3** College of Medicine, Kaduna State University, Kaduna, Kaduna State, Nigeria, **4** Barau Dikko Teaching Hospital, Kaduna, Kaduna State, Nigeria, **5** Department of Petroleum Chemistry, Baze University, Abuja, Nigeria

\* musakana77@yahoo.com

**Data Availability Statement:** Datasets and labels for variables in the datasets are available as supplementary files.

## Abstract

Despite the high burden of household air pollution from biomass fuel in sub-Saharan Africa, the association of prenatal biomass fuel exposure and birth weight as a continuous variable among term births has not been extensively studied. In this study, our primary aim is to estimate the association between biomass cooking fuel and birth weight among term births in Kaduna, northwestern Nigeria. For replication, we also evaluated this association in a larger and nationally representative sample from the 2018 Nigerian Demographic and Health Survey (DHS). Our primary analysis included 1,514 mother-child pairs recruited from Kaduna, in northwestern Nigeria, using the Child Electronic Growth Monitoring System (CEGROMS). Replication analysis was conducted using data from 6,975 mother-child pairs enrolled in 2018 Nigerian DHS. The outcome variable was birth weight, and the exposure was cooking fuel type, categorized in CEGROMS as liquefied petroleum gas, kerosene, or biomass fuel, and in the DHS as low pollution fuel, kerosene, or biomass fuel. We estimated covariate adjusted associations between birth weight and biomass fuel exposure in CEGROMS using linear regression and using linear mixed model in the DHS. In CEGROMS, adjusting for maternal age, education, parity, BMI at birth, and child sex, mothers exposed to biomass fuel gave birth to infants who were on average 113g lighter (95% CI −196 to −29), than those using liquified petroleum gas. In the 2018 Nigeria DHS data, compared to low pollution fuel users, mothers using biomass had infants weighing 50g (95% CI -103 to 2) lower at birth. Exposure to biomass cooking fuel was associated with lower birth weight in our study of term newborns in Kaduna, Nigeria. Data from the nationally representative DHS provide some support for these findings.

**Funding:** MAK, MS and SJL were supported by the Intramural Research Program of the National Institutes of Health (NIH), National Institute of Environmental Health Sciences, Z01 ES 49019. NIH had no role in study design and data collection, but as indicated NIH funded intramural investigators were involved in the analysis, decision to publish, and preparation of the manuscript. Epidemiological Resources and Investigation Consultancy (ERIC) Limited and Perinatal Epidemiological Research Unit (Kaduna, Nigeria) provided infrastructure, human resource, and logistics for data collection and management.

**Competing interests:** The authors have declared that no competing interests exist.

## Introduction

Biomass fuel, a significant source of indoor or household air pollution (IAP or HAP), includes wood, animal dung, charcoal, and crop residues and is used worldwide for cooking, heating, and lighting [1]. It is estimated that about 2.8 billion persons are exposed to HAP from biomass burning globally, with the bulk of the burden coming from Africa and Southeast Asia where over 60% of households cook with biomass fuel [2]. Particulate matter (PM2.5) and carbon monoxide (CO) levels have been observed to be higher in households using biomass fuel, and the exposures of adult women to these pollutants are substantially greater [3], mainly due to cooking activity and time spent indoors [4–7]. Consequently, prenatal biomass fuel exposure in women increases the risk of adverse birth outcomes [8, 9]. In a causal framework, elevated household $PM_{2.5}$ is related to reduced birth weight [3, 9, 10], which is, in turn, associated with neonatal and infant mortality, particularly in low- and middle-income countries [11, 12].

Mothers living in households that use biomass fuel have a 74% higher risk of giving birth to low birth weight infants than mothers who live in homes without air pollution from biomass fuel smoke [13]. Low birth weight occurs mainly due to preterm birth or intrauterine growth restriction whose aetiology is heterogenous [14]. Prenatal biomass exposure fits into a multifactorial model of the aetiology of low birth weight. During intrauterine life, it could affect foetal growth and development, directly through trans-placental exposure or indirectly resulting in a small-sized child at birth [15]. However, it is unclear if this reduced birth weight or size is due to preterm birth or intrauterine growth restriction, as few studies, including one from China [16], but none from sub-Saharan Africa (SSA) adjusted for gestational age. Because birth weight is highly influenced by gestational age [17, 18], studies limited to term infants can better estimate associations with birth weight independent of preterm birth. Also, shifts of birth weight distribution across the entire range [19], could be detected by assessing the linear association of biomass fuel exposure with birth weight as a continuous variable, but this has not been extensively investigated in SSA [9, 12, 20–22]. Furthermore, the precise effect estimate of biomass fuel alone on adverse birth outcomes, including LBW, has not been well characterized for SSA countries as most analyses of cross-sectional and household surveys use binary categories, low and high pollution (or unclean) fuels with both biomass fuel and kerosene grouped in the high category [12, 23].

Biomass cooking fuel is commonly used in Nigeria [23], the most populous in Africa, and among the five countries where most preterm and small-for-gestational-age infants are born globally [24]. Contemporary studies investigating the association of biomass fuel and adverse birth outcomes in the country are few and those exploring the specific impact of biomass fuel on birth weight are fewer [23, 25]. It has been documented that birth weight is a crucial indicator of foetal and neonatal health [26]. In the Nigerian context, adverse birth outcomes, including low birth weight, are known to have regional variation, with the burden significantly more in the northern parts [23, 27]. Presently, it is unclear if the observed regional inequalities apply in the association of cooking fuel types with birth weight as a continuous variable among term births. Therefore, in this study, our primary aim is to determine the association between biomass fuel and birth weight among term births in Kaduna, northwestern Nigeria. Our secondary aim is to attempt to replicate findings in a more extensive and nationally representative sample from 2018 Nigerian Demographic and Health Survey (DHS).

## Methods

### Study setting and populations

Our primary analysis is based on a cross-sectional study conducted at the Child Welfare Clinic of Barau Dikko Teaching Hospital, Kaduna (Kaduna State, northwestern Nigeria). The

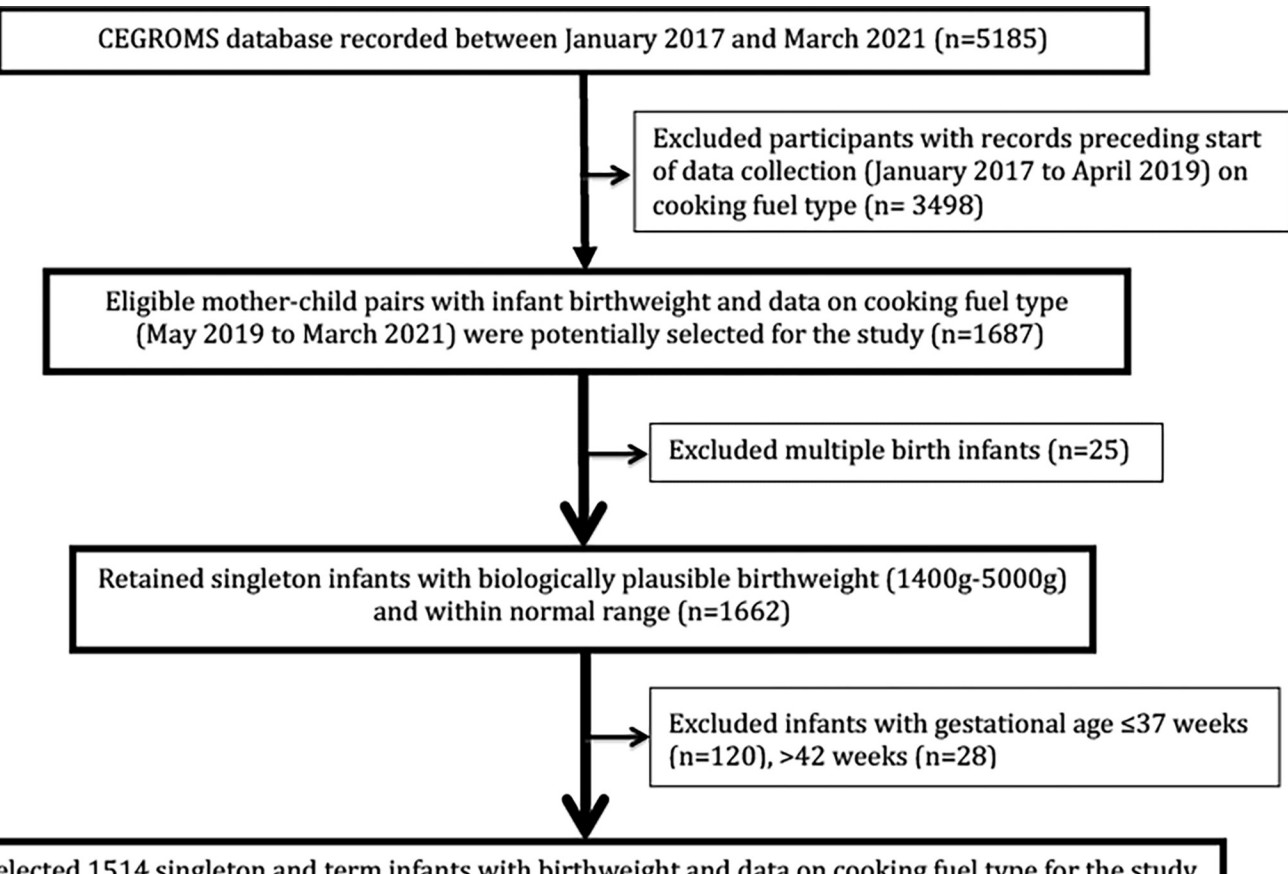

**Fig 1. Participant selection flowchart from the CEGROMS.**

catchment area of this hospital is the entire city of Kaduna, the fourth largest in Nigeria with a population of 1.6 million. Data for this study was obtained from the Child Electronic Growth Monitoring System (CEGROMS) as has been described previously [28]. The larger CEGROMS population included all women from the catchment area who brought their newborns to the study site for BCG vaccination and growth monitoring within the first week of life over the period of January 2017–March 2021. In this analysis, we included only mother-child pairs recruited during the period when we collected data on household cooking fuel type (May 2019 to March 2021). The study population consisted of 1,514 mothers with singleton births and complete data on gestational age at birth, birth weight and cooking fuel as outlined in the participant selection flowchart (Fig 1). Multiple births were excluded because of their higher risk of preterm birth and LBW.

Our replication analysis was performed using data from the 2018 Nigeria Demographic and Health Survey (DHS), which is the is the most recent available Nigerian DHS data and contained the variable that measured gestational age as duration of pregnancy [29]. The DHS is a nationally representative survey using a multistage stratified probabilistic sampling design including sampling weights conducted approximately every five years in many low and lower-middle income countries. The DHS collects data on multiple indicators of health and social and demographic characteristics as well data on reproductive life. The 2018 Nigeria DHS encompassed a total of 33,924 women with childbirth during the last 5 years prior to the

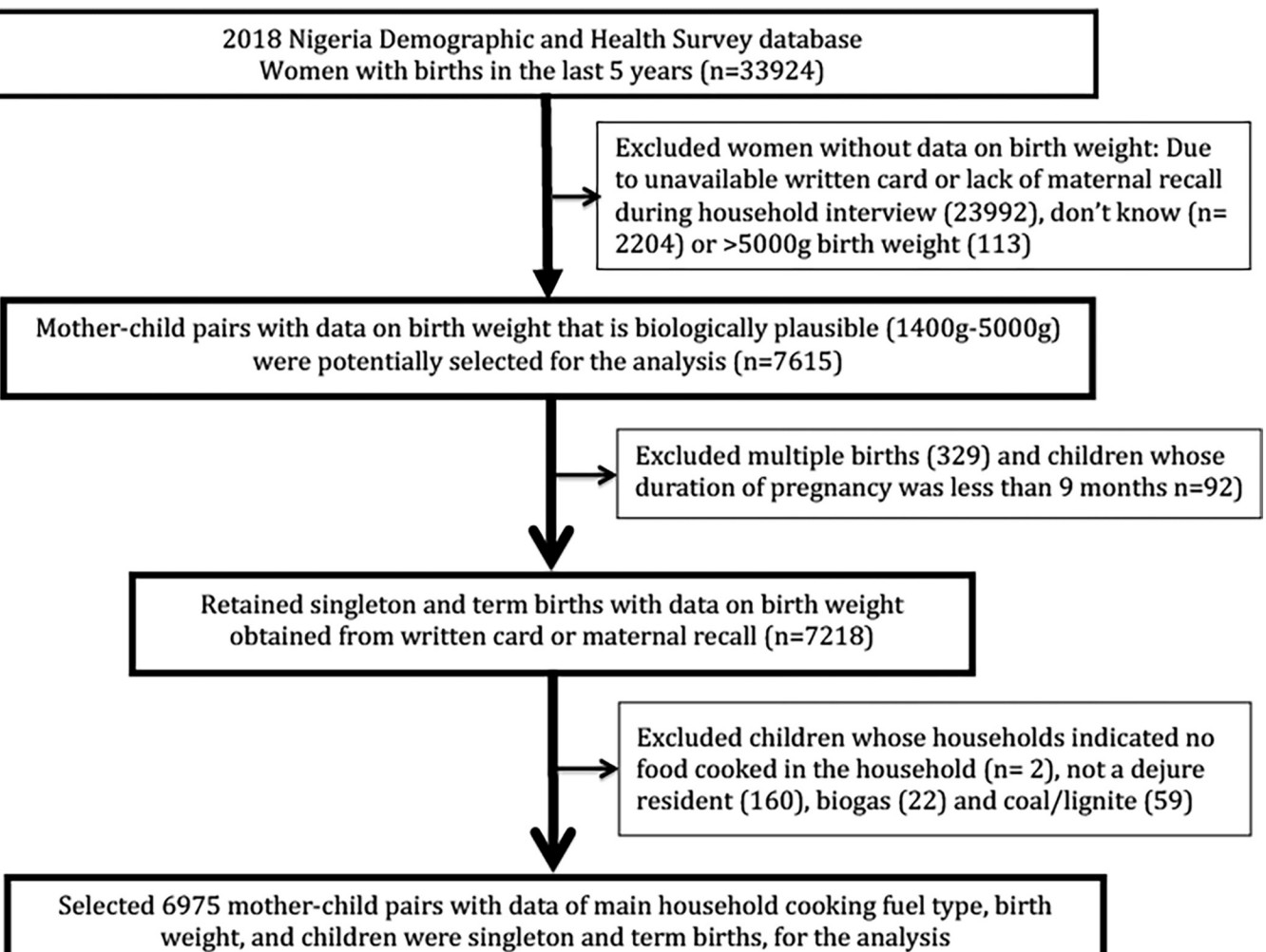

**Fig 2. Participant selection flowchart from the 2018 Nigerian DHS.**

survey. The final sample size for this study is 6,975 (Fig 2) comprising of women with complete data on duration of pregnancy, birth weight, and main cooking fuel type (S1 Table). Multiple births (twins), children whose households responded "other" or "no food cooked in the household" to the type of cooking fuel used, were excluded from the analysis.

## Ethical consideration

Institutional Review Board approval was obtained from the Health Research Ethics Committee of the Ministry of Health and Human Services, Kaduna State, Nigeria (MOH/ADM/744/VOL/584). All the participants provided written informed consent by thumb printing. Participants were assured that the study was anonymous, and their privacy and confidentiality protected by the removal of identifying information during all the stages of data management, analysis, and dissemination (S1–S4 Data). Study data were collected and managed using Research Electronic Data Capture (REDCap) tools hosted at Perinatal Epidemiological Research Unit, Kaduna, Nigeria.

### Exposure and outcome measurements

**Outcome variables.** Birth weight (in grams) in CEGROMS was recorded as a continuous variable on the child health (immunisation and growth monitoring) card. Trained nurse-midwives measured the birth weight using a digital scale at birth for infants delivered in the health facility or within 48 hours, during the BCG immunisation visit, for those delivered at home. Birth weight was measured without clothing. Scales were calibrated using a standard weight to ensure accuracy and comparability. We restricted the analysis to term births (≥37 completed weeks). The algorithm for identifying term births was based on self-report of the last menstrual period (LMP). We had imprecision in the estimation of gestational age because some women did not remember their LMP and women in our population generally report their own gestational duration in whole completed months rather than in weeks. There are approximately 4.25 weeks in a month and through the conversion of months to weeks for data entry, 9-month pregnancies were recorded as 38.25 weeks (9 months times 4.25 weeks).

For the DHS, the outcome variable was birth weight (in grams) measured at birth. During the interview, mothers were asked if their child was weighed at birth and if they had a health card of the child's birth weight record. If not, mothers were asked, if possible, to recall the weight of their child at birth. Measured or recalled birth weight (S2 Table) was used as a continuous variable in the analysis. The gestational age at birth was measured as the duration of pregnancy and recorded as the number of completed months or weeks. To harmonize gestational age across the data sources, and in consultation with clinicians and data entry personnel in Nigeria, the following procedures were used. The number of weeks was multiplied by 0.23 to convert to the number of months and then rounded down to the nearest whole number to get the number of completed months.

**Exposure variables.** In the CEGROMS dataset, mothers selected their primary type of household cooking fuel during pregnancy from the following options: liquified petroleum gas (LPG), kerosene, charcoal, wood, crops or straw, and animal dung. For analytical purposes, we classified cooking fuel types as LPG, kerosene, and biomass fuel (including charcoal, wood, crops or straw, and animal dung). For the DHS dataset, maternal exposure to biomass fuels was determined through the DHS household questionnaire. During the interview, mothers were asked about their current household cooking fuel type at varying postnatal periods. The respondents could choose among the following cooking fuel types: Electricity, LPG, natural gas (NG), biogas, kerosene, coal/lignite, charcoal, wood, straw/ shrubs/grass, agricultural crop, animal dung, and no food cooked in the house. In DHS, we categorized the types of cooking fuels into low pollution fuel (Electricity, LPG, and NG), kerosene, and biomass fuel (charcoal, wood, crops, straw, and dung). We excluded women who used biogas (N = 26) and coal/lignite (N = 59) because of their small sample size, with no users of either cooking fuel type in three of the six geopolitical regions of the country.

**Covariates.** We selected covariates as potential confounders because of their significant association with birth weight (S3 Table) documented in the literature [14]. The following covariates were included in the CEGROMS analysis: maternal age classified as <20, 20–34, and ≥35 years, birth order of the index child categorized as 1, 2, 3 and 4+, maternal educational level as a categorical variable (none, primary, secondary, and tertiary), and occupational status was reported and categorized according to the International Standard Classification of Occupations (ISCO-08) [30]. The classes were professionals and managers,: technicians and associate professionals, clerks, service and sales workers, skilled agricultural and fishery workers, craft and related trades, plant and machine operators and assemblers, elementary occupations; or unemployed. The remaining covariates were maternal body mass index (BMI) calculated from weight and height measurements during the BCG immunization, the number

of antenatal care visits (1–4 or >4 times), place of delivery (health facility or home), type of delivery (vaginal or caesarean section), and child sex.

In the analysis of the 2018 DHS dataset, we included the following hierarchy of covariates as categorical variables (S4 Table): region (north-central, north-east, south-east, south-south, and south-west), type of place of residence (urban or rural), number of household members (<5, 5–9 and ≥10), and wealth index categories based on item count of household ownership of assets (≤6, 7–12 and 13–24). Because cooking fuel type is the main exposure variable for our study, we reconstructed the wealth index by excluding cooking fuel from the Demographic and Health Survey household possession items (count-based) used to derive the wealth index [31]. Our reclassified wealth index and the original wealth index were comparable. Other covariates included maternal age at birth of the child (<20, 20–34 and ≥35 years), educational level (none, primary, secondary, and higher), birth order or parity (1, 2, 3 and 4 or above), maternal smoking (yes and no), number of antenatal care visits (1–4 and >4 times), place of delivery (health facility and home), type of delivery (caesarean section and vaginal delivery), and child sex.

## Statistical analysis

We performed descriptive analysis of maternal and infant characteristics by household cooking fuel type among term births in the CEGROMS and 2018 Nigeria DHS datasets. ANOVA was used to compare continuous variables, while comparison of categorical variables was done with chi squared and Fischer's exact tests. In our primary analysis using CEGROMS, we performed linear regression to test for the relationship of cooking fuel type as a 3-category variable (LPG, kerosene, and biomass fuel) or as a 2-category variable (either LPG or kerosene, biomass fuel) and birth weight among term births. We estimated the β (95% Confidence Interval) coefficient as the difference in birth weight (in grams) for exposure to biomass fuel or kerosene compared to LPG (reference category) in the analysis using the 3-category cooking fuel variable. LPG or kerosene is the reference category in the analysis using the 2-category cooking fuel variable. Likelihood ratio tests were done to test the difference between the model with 3-category cooking fuel variable vs the 2-category cooking fuel variable. Because we did not observe any difference between the two models, we used the 2-category fuel variable to test for interaction effects with all the adjusted factors. Each model has all the covariates, but the interaction terms were tested one at a time in different models. In all models, we adjusted for maternal age, birth order (parity), body mass index, child sex, and maternal education as a proxy for socio-economic status. Least-square means, which are the predicted marginal means, were computed. We did not adjust for maternal occupational status because it is corelated with education.

We performed stratified analysis to compare the association between biomass fuel and birth weight across different categories of the covariates. We used an interaction term between cooking fuel type (2 category; LPG/kerosene or biomass fuel) and each of the selected covariates to determine if the associations between biomass fuel with birth weight significantly (interaction p-value <0.05) differed between the categories of the selected covariates.

We attempted to replicate our primary analysis using a dataset from the 2018 Nigeria DHS. Considering that some mothers have multiple children represented in the dataset, and the effects of clustering at family/ household and community levels, we applied linear mixed effects with random intercepts for cluster ID and mother's unique ID (nested within cluster ID). The 3-category cooking fuel variable (low pollution fuel, kerosene, and biomass fuel) and 2-category cooking fuel variable (low pollution fuel/kerosene and biomass fuel) were the exposure variables. Adopting the same analytical strategy as our primary analysis using the CEGROMS

dataset, we estimated the β (95% CI) coefficient as the difference in birth weight (in grams) per exposure of biomass fuel or kerosene compared to low pollution fuel (3-category cooking fuel variable) and low pollution fuel/kerosene (2-category cooking fuel variable). Least-square means, which are the predicted marginal means, were computed. We adjusted for the community level factors (region and type of residence), household level factors (number of household members and wealth index) and individual level factors (maternal age, education, type of delivery, birth order and child sex). Stratified analysis and estimation of interaction (interaction p-value 0.05 considered significant) was performed to compare the association between biomass fuel and birth weight for the various categories of the identified covariates. We set the threshold of statistical significance for all analyses at 0.05. SPSS V.23 and R version 4.0.2 for Windows were used to perform the analyses.

## Results

### Association between biomass cooking fuel and birth weight among term births in the Child Electronic Growth Monitoring System (CEGROMS)

Socio-demographic characteristics of the CEGROMS mother-child population by cooking fuel type are presented in Table 1. Mothers using the three types of cooking fuels differed by several socio-demographic factors. Compared to LPG and kerosene users, mothers using biomass fuel were less educated and fewer had a professional/managerial occupation. The prevalence of biomass use increased with increasing parity but decreased with increasing BMI. There was no difference by child sex. In this univariate comparison, birth weight was lower in infants of mothers who were using biomass.

Using the 3-category cooking fuel variable, on average, infants of mothers exposed to biomass fuel were 113g lighter (95% CI −196 to −29), compared with those of mothers using LPG, adjusting for maternal age, education, parity, BMI at birth, and child sex (Table 2). There was no appreciable difference in birth weight of infants of mothers using kerosene compared with mothers using LPG. The likelihood ratio test for a model with 3-category fuel variable compared to a 2-category fuel variable had a p-value of 0.40. Therefore, we used the 2-category fuel variable to test for interaction effects with other variables (Table 3). The beta estimates in Table 3 are the effects of biomass in the corresponding strata of the variables. Only one of these gave a P value for interaction <0.05; the reduction in birth weight associated with biomass fuel increased across categories of maternal normal weight, overweight and obese ($P_{interaction}$ = 0.04).

### Association between biomass cooking fuel and birth weight among term births: Results from the 2018 Nigeria Demographic and Health Survey

Births from the same cluster were correlated with an intraclass correlation coefficient (ICC) of 0.05 (likelihood ratio test $\chi^2$ = 31.9). As expected, different births from the same woman were highly correlated with an ICC of 0.42 (likelihood ratio test $\chi^2$ = 249.1) as determined by the three-level linear mixed effect model. In Table 4, we describe the socio-demographic characteristics of the mother-child pairs included in our replication analysis based on the 2018 Nigeria Demographic and Health Survey data. Women using the three types of cooking fuels differed by community, household/family, and individual level factors. The proportion of mothers using biomass fuel was greater in the northern parts of the country. The prevalence of biomass use increased with increasing number of people living in a household but reduced with increasing wealth index. The biomass fuel users were less educated compared with low

**Table 1. Maternal and infant characteristics by cooking fuel type among term births, CEGROMS (N = 1514).**

| Variable, n (%) | LPG* | Kerosene | Biomass | P-value‡ |
|---|---|---|---|---|
| n (%) | 1194 (78.9) | 134 (8.9) | 186 (12.3) | |
| **Maternal age** | | | | |
| <20 years | 36 (75.0) | 5 (10.4) | 7 (14.6) | <0.01 |
| 20–34 years | 934(81.1) | 93 (8.1) | 124 (10.8) | |
| >34 years | 224 (71.1) | 36 (11.4) | 55 (17.5) | |
| **Birth order (parity)** | | | | |
| 1 | 385 (89.7) | 24 (5.6) | 20 (4.7) | <0.01 |
| 2 | 322 (85.6) | 28 (7.4) | 26 (6.9) | |
| 3 | 219 (79.6) | 33 (12.0) | 23 (8.4) | |
| 4+ | 268 (61.8) | 49 (11.3) | 117 (27.0) | |
| **Maternal educational level** | | | | |
| None | 3 (13.0) | 1 (4.3) | 19 (82.6) | <0.01 |
| Primary | 21 (28.4) | 11 (14.9) | 42 (56.8) | |
| Secondary | 446 (70.2) | 81 (12.8) | 108 (17.0) | |
| Higher | 724 (92.6) | 41 (5.2) | 17 (2.2) | |
| **Maternal occupational status**** | | | | |
| Professional and managers | 287 (92.9) | 15 (4.9) | 7 (2.3) | <0.01 |
| Technicians and associate professionals | 278 (74.5) | 42 (11.3) | 53 (14.2) | |
| Unemployed/housewife | 629 (75.6) | 77 (9.3) | 126 (15.1) | |
| **Maternal BMI** | | | | |
| Normal weight (<25 kg/m²) | 478 (74.2) | 60 (9.3) | 106 (16.5) | <0.01 |
| Overweight (25.0–29.9 kg/m²) | 393 (79.2) | 47 (9.5) | 56 (11.3) | |
| Obese (≥30 kg/m²) | 323 (86.4) | 27 (7.2) | 24 (6.4) | |
| **Number of antenatal care visits** | | | | |
| 1–4 visits | 345 (74.4) | 42 (9.1) | 77 (16.6) | 0.01 |
| >4 visits | 848 (80.9) | 92 (8.8) | 108 (10.3) | |
| **Place of delivery** | | | | |
| Health facility | 1163 (80.9) | 122 (8.5) | 153 (10.6) | <0.01 |
| Home | 31 (40.8) | 12 (15.8) | 33 (43.4) | |
| **Delivery method** | | | | |
| Vaginal | 1067 (77.7) | 127 (9.2) | 180 (13.1) | <0.01 |
| Caesarean section | 127 (90.7) | 7 (5.0) | 6 (4.3) | |
| **Child sex** | | | | |
| Male | 622 (79.4) | 59 (7.5) | 102 (13.0) | 0.12 |
| Female | 572 (78.2) | 75 (10.3) | 84 (11.5) | |
| **Birth weight (g),** mean (sd) | 3171 (485) | 3128 (397) | 3059 (449) | 0.01 |

*LPG = Liquified petroleum gas

**ISCO-08 = Technicians and associated professionals include clerks, service and sales workers, skilled agricultural and fishery workers, craft and related trades, Plant and machine operators and assemblers, elementary occupations.

‡P-value for difference in proportions within categories of maternal and infant characteristics between women using different cooking fuel types.

pollution fuel users. In this univariate comparison of the cooking fuel types, birth weight was lower in infants of women who were using biomass.

The association between cooking fuel type (3-category variable) and birth weight in the DHS adjusted for the community, household/ family, and individual-level factors is shown in Table 5. Compared to low pollution fuel users, mothers using biomass had infants weighing

**Table 2. Association between cooking fuels and birth weight (in grams) among term births, result of linear regression from CEGROMS (N = 1514).**

| Predictor | N | Least square mean birth weight, g (SE) | β (95% confidence Interval) |
|---|---|---|---|
| Liquified petroleum gas (LPG) | 1194 | 3210 (40) | Reference |
| Kerosene | 134 | 3173 (53) | -36.6 (-121.1 to 48.0) |
| Biomass fuel | 186 | 3097 (45) | -112.5 (-195.7, -29.2) |

*β = difference in birth weight (in grams) for exposure to kerosene and biomass fuel relative to LPG. Adjusted for maternal age, birth order (parity), educational level, BMI, and child sex. CI = Confidence interval.

50g (95% CI -103 to 2) less at birth. The birth weight of infants of mothers from households using kerosene did not differ appreciably or statistically significantly from that of infants of mothers using low pollution fuel (14g lower, 95% CI -67 to 38). We evaluated interaction with the potential confounders using the 2-category fuel type variable because it also was comparable with the 3-category variable model (likelihood ratio test p-value = 0.59). The beta estimates in Table 6 are the effects of biomass in the corresponding strata of the variables. The P value

**Table 3. Associations between biomass fuel and birth weight by levels of other factors in CEGROMS (N = 1514).**

| Variable | β (95% confidence Interval)* | Interaction P-value‡ |
|---|---|---|
| **Biomass vs LPG and kerosene (no interaction model)** | -114.7 (-196.3 to -33.2) | |
| **Maternal age, years** | | |
| <20 years | -90.8 (-461.7 to 280.2) | 0.96 |
| 20–34 years | -99.4 (-195.2 to -3.6) | |
| ≥35 years | -123 (-263.8 to 17.8) | |
| **Birth order (parity)** | | |
| 1 | -56.9 (-268.2 to 154.5) | 0.88 |
| 2 | -78.8 (-265.1 to 107.5) | |
| 3 | -74.8 (-276.5 to 126.9) | |
| 4+ | -132.4 (-237.2 to -27.7) | |
| **Maternal education** | | |
| Higher | -195.7 (-418.8 to 27.3) | 0.19 |
| Secondary | -61.2 (-158.2 to 35.8) | |
| Primary | -161.8 (-374.3 to 50.6) | |
| No education | -547.7 (-1046 to -49.3) | |
| **Body mass index** | | |
| Normal weight (<25 kg/m$^2$) | -33.9 (-139.3 to 71.6) | 0.04 |
| Overweight (25.0–29.9 kg/m$^2$) | -138.5 (-271.7 to -5.4) | |
| Obese (≥30 kg/m$^2$) | -309 (-504.5, -113.6) | |
| **Child sex** | | |
| Male | -74.5 (-178 to 29) | 0.34 |
| Female | -144.4 (-258.2 to -30.7) | |

The reference category is Liquified Petroleum Gas (LPG) and kerosene.

*β = Difference in birth weight (in grams) for exposure to biomass fuel relative to LPG and kerosene by category of maternal age, birth order (parity), maternal educational level, body mass index and child sex. From linear regression with adjustment for maternal age, parity, educational level), maternal body mass index and child sex.

CI = Confidence interval.

‡**Interaction P-value**: The interaction effects were tested using likelihood-ratio test. The variables were treated as categorical and the degrees of freedom of each test are number of levels of each variable minus one.

**Table 4. Maternal and infant characteristics by cooking fuel type among term births, 2018 Nigeria Demographic and Health Survey (N = 6975)\*.**

| Characteristics | Low-pollution fuel | Kerosene | Biomass fuel | P-value |
|---|---|---|---|---|
| n | 1615 (23.2) | 1411 (20.2) | 3949 (56.6) | |
| **Region** | | | | |
| North-central | 237 (16.5) | 190 (13.3) | 1006 (70.2) | <0.01 |
| North-west | 138 (24.2) | 33 (5.8) | 400 (70.1) | |
| North-east | 20 (3.9) | 17 (3.3) | 472 (92.7) | |
| South-east | 174 (9.9) | 455 (25.9) | 1129 (64.2) | |
| South-south | 293 (27.3) | 327 (30.4) | 454 (42.3) | |
| South-west | 753 (46.2) | 389 (23.9) | 488 (29.9) | |
| **Type of place of residence** | | | | |
| Urban | 1363 (32.1) | 1074 (25.3) | 1809 (42.6) | <0.01 |
| Rural | 252 (9.2) | 337 (12.3) | 2140 (78.4) | |
| **Number of household members,** | | | | |
| <5 | 781 (31.0) | 585 (23.2) | 1156 (45.8) | <0.01 |
| 5–9 | 582 (23.2) | 522 (20.8) | 1404 (56.0) | |
| ≥10 | 252 (13.0) | 304 (15.6) | 1389 (71.4) | |
| **Wealth index (Item count)** | | | | |
| ≥6 | 8 (0.9) | 32 (3.9) | 810 (95.3) | <0.01 |
| 7–12 | 257 (10.2) | 492 (19.5) | 1774 (70.3) | |
| 13–24 | 1350 (37.5) | 887 (24.6) | 1365 (37.9) | |
| **Maternal age (years)** | | | | |
| <20 years | 30 (6.9) | 63 (14.5) | 342 (78.6) | <0.01 |
| 20–34 years | 1296 (24.0) | 1118 (20.7) | 2983 (55.3) | |
| >34 years | 289 (25.3) | 230 (20.1) | 624 (54.6) | |
| **Maternal education** | | | | |
| None | 15 (2.6) | 37 (6.4) | 525 (91.0) | <0.01 |
| Primary | 56 (6.2) | 132 (14.6) | 718 (79.2) | |
| Secondary | 677 (17.9) | 903 (23.9) | 2198 (58.2) | |
| Higher | 867 (50.6) | 339 (19.8) | 508 (29.6) | |
| **Maternal smoking** | | | | |
| No | 1609 (23.1) | 1410 (20.3) | 3940 (56.6) | 0.23 |
| Yes | 6 (37.5) | 1 (6.3) | 9 (56.3) | |
| **Birth order (parity)** | | | | |
| 1 | 514 (28.6) | 387 (21.5) | 897 (49.9) | <0.01 |
| 2 | 466 (29.6) | 344 (21.9) | 763 (48.5) | |
| 3 | 299 (23.0) | 282 (21.7) | 721 (55.4) | |
| 4+ | 336 (14.6) | 398 (17.3) | 1568 (68.1) | |
| **Type of delivery** | | | | |
| Caesarean section | 214 (43.4) | 112 (22.7) | 167 (33.9) | <0.01 |
| Vaginal | 1389 (21.7) | 1269 (19.8) | 3751 (58.5) | |
| **Child sex** | | | | |
| Male | 863 (24.2) | 725 (20.3) | 1981 (55.5) | 0.08 |
| Female | 752 (22.1) | 686 (20.1) | 1968 (57.8) | |
| **Birth weight (g)** mean (SD) | 3328 (61.5) | 3347 (64.3) | 3255 (65.0) | <0.01 |
| **Source of information about birth weight** | | | | |
| From written card | 557 (20.9) | 480 (18.0) | 1631 (61.1) | <0.01 |

*(Continued)*

**Table 4.** (Continued)

| Characteristics | Low-pollution fuel | Kerosene | Biomass fuel | P-value |
|---|---|---|---|---|
| From mother's recall | 1058 (24.6) | 931 (21.6) | 2318 (53.8) | |

*Dataset of children of women that gave birth in the last 5 years with data on birth weight and cooking fuel type.

P-value = Difference in proportions of categories of maternal and infant characteristics between women using different cooking fuel types.

for interaction was less than 0.05 only for infant sex, with maternal biomass fuel use significantly related to reduced birth weight only among boys. We observed a regional difference in this interaction between biomass fuel and infant sex (S5 Table); there was no evidence that the association differed by sex in the northwest region (Pinteraction = 0.57), where the CEGROMS participants were enrolled.

## Discussion

This study used two data sources to investigate the association between biomass fuel and birth weight among term birth infants in the Nigerian population. Our primary analysis consisted of a sample of mother-child pairs recruited from Kaduna, northwest Nigeria (CEGROMS). The findings supported the hypothesis that maternal use of biomass fuel is associated with reduced birth weight, as a continuous variable, among term births. We attempted to replicate these findings using data from 2018 Nigerian Demographic and Health Survey (DHS), which produced results similar in direction but of lesser magnitude.

Population-level evidence of the relationship of maternal exposure to biomass fuel and reduced birth weight in SSA is mainly from national surveys that previously lacked data on the duration of pregnancy [12, 32–34]. Therefore, we took advantage of inclusion of data on duration of pregnancy in the latest Nigerian DHS to evaluate the linear relationship of biomass fuel exposure and birth weight among term newborns [18]. Our findings are consistent with previous studies reporting a negative association between biomass fuel exposure and birth weight as a continuous variable for all births unstratified by gestational age categories [9, 16, 20, 21, 35]. Studies that examined this association in term births are scarce in Africa. In a Chinese birth cohort study that examined the association of biomass fuel and birth weight as a binary variable (low weight versus normal) a significant negative association was seen only in preterm births [16]. However, power was limited for analysis of biomass among term births with only 7 exposed low birth weight babies [16]. We infer that differences in analytical design and population could be the reasons for these divergent observations.

**Table 5. Association between birth weight among term births and maternal use of cooking fuel type in the, 2018 Nigeria Demographic and Health Survey (N = 6975).**

| Predictor | N | Least square mean birth weight, g (SE) | β (95% confidence Interval) |
|---|---|---|---|
| Low pollution fuel (LPF) | 1615 | 3291 (29) | Reference |
| Kerosene | 1411 | 3276 (28) | -14.4 (-66.6 to 37.7) |
| Biomass fuel | 3949 | 3240 (21) | -50.2 (-102.6, 2.1) |

The reference category is Low Pollution Fuel (LPF). BW = Birth weight.

*β = Difference in BW (in grams) for exposure to a kerosene and biomass fuel relative to low pollution fuel from a mixed linear regression model. Adjusted for the hierarchy of covariates: Region, type of residence, number of household members, maternal age, birth order (parity), maternal education, wealth index, child sex and delivery method.

**Table 6. Associations between biomass fuel and birth weight by levels of other factors in the 2018 Nigeria Demographic and Health Survey (N = 6975).**

| Variable | B* (95% confidence Interval) | Interaction P-value‡ |
|---|---|---|
| **Biomass vs LPF and kerosene (no interaction model)** | -41.8 (-84.4 to 0.8) | |
| **Region** | | |
| Northcentral | -39 (-124.1 to 46.0) | 0.16 |
| Northwest | -105.9 (-238 to 26.2) | |
| Northeast | -232.2 (-456.2 to -8.2) | |
| Southeast | -2.7 (-77 to 71.6) | |
| South-south | -105.1 (-194.9 to -15.3) | |
| South West | -4.4 (-82.2 to 73.4) | |
| **Type of residence** | | |
| Urban | -56.2 (-105.2 to -7.1) | 0.25 |
| Rural | -9.3 (-78.8 to 60.2) | |
| **Number of household members** | | |
| <5 | 0.2 (-58.5 to 58.9) | 0.11 |
| 5–9 | -62.2 (-122.2 to -2.2) | |
| ≥10 | -81.3 (-155 to -7.6) | |
| **Maternal age, years** | | |
| <20 years | -36.5 (-181.8 to 108.7) | 0.15 |
| 20–34 years | -56.8 (-102.4 to -11.1) | |
| ≥35 years | 23.5 (-55.4 to 102.5) | |
| **Birth order (parity)** | | |
| 1 | -39.2 (-103 to 24.6) | 0.69 |
| 2 | -63.7 (-129.9 to 2.4) | |
| 3 | -59.3 (-130.1 to 11.6) | |
| 4+ | -18.7 (-81.3 to 43.8) | |
| **Maternal education** | | |
| No education | -89.7 (-284.2 to104.7) | 0.38 |
| Primary | -53.5 (-163.2 to 56.2) | |
| Secondary | -19.9 (-69.9 to 30.1) | |
| Higher | -89.8 (-165 to -14.6) | |
| **Wealth index** | | |
| ≥6 | -103.2 (-310.7 to 104.4) | 0.61 |
| 7–12 | -58.1 (-119.8 to 3.6) | |
| 13–24 | -27.7 (-80.1 to 24.8) | |
| **Child sex** | | |
| Male | -70.7 (-121.2 to -20.3) | 0.04 |
| Female | -10.3 (-62.1 to 41.5) | |
| **Delivery method** | | |
| Vaginal | -43.3 (-86.7 to 0.2) | 0.74 |
| Cesarean section | -22.4 (-143.6 to 98.9) | |

BW = Birth weight.

*β = Difference in BW (in grams) for exposure to a type of cooking fuel relative to low pollution fuel (LPF) and kerosene.

‡**Interaction P-value:** The interaction effects were tested using likelihood-ratio test. The variables were treated as categorical and the degrees of freedom of each test degrees of freedom of each test are number of levels of each variable minus one.

Mothers living in the Northern states of Nigeria, where Kaduna is located, are known to have a higher risk of adverse birth outcomes [23]. Our estimate of the effect of biomass fuel on birth weight as a continuous variable was higher in the infants from the Kaduna study (CEGROMS) than the nationally representative sample of the 2018 Nigerian DHS. Differences in methodological approaches could explain the variation between the Kaduna study and replication analysis. The CEGROMS exposure and outcome data were collected postnatally in the first week of life by asking mothers directly about the cooking fuel type used during the index pregnancy and recording the infant birth weight. Conversely, in the DHS, eligible women were asked about their current household cooking fuel type at varying postnatal periods. Thus, information regarding fuel use when pregnant with the index child may be less precisely recalled. In addition, for a substantial proportion of mothers in the DHS infant birth weight data were missing either because the health card was not available, or the mother could not recall. Birth weight data were not missing at random. Rather, missing birth weight data were higher in lower socio-economic level households where biomass fuel use is more common, which could attenuate the observed association between biomass fuel and birth weight. We acknowledge that in the DHS analysis stratified by region, the association of biomass fuel use and birth weight was not significant in northwest Nigeria, the location of Kaduna. But it was a bit stronger than the association in three of the six regions. However, across the six geopolitical regions in Nigeria, we also observed that infant birth weight was lowest in northwest Nigeria, which might be due to the regional differences in the predictors of birth weight.

Achieving a national reduction of adverse birth outcomes like low birth weight will depend on addressing the regional disparities in the key determinants [23, 27]. Nigeria is among the sub-Saharan African countries with the highest fertility rate in the world [36], and higher household size (number of people, a proxy for population density) has been linked to poorer maternal and child health outcomes [37]. Our results showed that parity, household size, and wealth index influence cooking fuel choice which is consistent with existing literature [38]. It is reasonable that multiparous women, living in large-sized (five or more persons) or poorer households will disproportionally lack access to cleaner cooking fuels. This could be due to cost, mismatches between cooking technologies and household needs, and unreliable fuel supply [39]. Consequently, women with more children or living in large-sized households might cook with bigger pots, requiring longer cooking times, which increases their exposure levels of indoor air pollution from biomass fuel use.

To mitigate the health effects of IAP, the World Health Organization recommends a shift to cleaner fuels rather than the promotion of technologies that more efficiently combust biomass fuels [40]. Presently, biomass fuels and kerosene are still widely used in urban and rural Nigerian communities due to supply and demand issues driving household energy choices [41]. Biomass fuel and kerosene are often jointly classified as high pollution fuels that adversely affect birth weight [21, 23, 42], despite each cooking fuel type having a separate impact on birth outcomes [43]. In our primary and replication analyses, we noted that compared to mothers using cleaner fuels, the effect estimates of the association between cooking fuel and infant birth weight are greater for biomass fuel than kerosene. Epidemiological studies that investigated the separate impact of biomass fuel and kerosene exposure on birth weight are lacking. In an Indian population study, biomass fuel and kerosene exposure both were associated with significantly decreased birth weight, though kerosene exposure had a larger effect than biomass fuel [43]. More research is needed to clarify the specific aetiological relationship of biomass or kerosene exposure and birth weight in the African population. In the future, we hope to implement an objective measurement of IAP to characterize the local exposure and to conduct a birth cohort study in Kaduna to explore the association with birth weight in more detail.

Our study highlights the effect of biomass fuel on birth weight among term births in a Nigerian population and the need for public health intervention to mitigate it. We recommend that pregnant women attending antenatal care be asked about cooking fuels and provided adequate resources to minimize prenatal biomass exposures. Some recent intervention trials conducted in low- and medium-income countries, including Nigeria, have shown that transition from biomass fuel to cleaner fuels reduces household $PM_{2.5}$ and CO [44], and improves births outcomes like birth weight [25, 45]. However, scaling up interventions entails understanding the determinants of household cooking fuel use to strengthen the effectiveness of interventions, broaden coverage, and make adoption of cleaner fuels in the general population sustainable [46]. Actionable strategies for reducing the presence of indoor pollutants and personal exposures could be developed by combining scientific knowledge about the effectiveness of existing interventions and a social-ecological systems framework: synthesis of existing interventions and literature to elucidate relationships among spatially and otherwise diverse indoor air quality factors [47]. Specific policy instruments intended to reduce population-level exposure to indoor air pollution from burning biomass fuel include stove subsidy, fuel subsidy, fuel bans, and behavior change communication [48]. Finally, it is vital to adopt the lessons from similar household health interventions, such as sanitation and nutrition, in implementing changes in the energy ecosystems to support the scaling of clean fuels. This can be accomplished by intrinsically involving key institutional actors outside the health sector and using existing implementation science frameworks to enhance understanding barriers to and enablers of adoption [49].

## Strengths and limitation

Exposure to biomass fuel was assessed postnatally. However, we expect negligible recall bias because household cooking fuel type might not vary over the short period from pregnancy to very early infancy. The way we calculated gestational age using months based on maternal self-report after birth is a limitation because of misclassification from imprecise reporting. The difficulty of enrolling women delivering at home is a major challenge to getting a representative sample of mothers for perinatal research in Nigeria as we found in CEGROMS [28]. Our replication analysis with DHS data addressed this limitation as the survey recruited a weighted sample of mothers with adequate representation of those who delivered at home. All the infants in the CEGROMS dataset had measured birth weight, but there is potential bias in birth weight measurement within 48 hours for infants born at home [19], as it is known that there is a 5%–10% loss of birth weight during the initial few days of life [50, 51]. However, the proportion of mothers that gave birth at home in the CEGROMS dataset was too low to influence the mean birth weight [28]. There could be unmeasured confounding for factors we did not assess, especially hypertension in pregnancy and pregnancy-associated malaria. However, we had a period of recruitment covering more than a year, which should standardize the seasonal exposure to malaria [52, 53].

Our study has many strengths including the large sample size of CEGROMS dataset relative to some previous studies [54, 55]. Additionally, we replicated our findings using a larger dataset from the 2018 Nigerian DHS, which yielded results of comparable direction although of slightly lower magnitude. The DHS data is characterized by multilevel factors with potentially clustering effects at community and or household/family levels, which we addressed with linear mixed (multilevel or hierarchical) regression analysis. We derived a new household wealth index variable for use in our analysis in the DHS because cooking fuel is a principal component of the standard wealth index but is also the primary exposure of interest in our analysis. Maternal smoking was negligible in the CEGROMS and DHS population, and adjusting for it did not make much difference on the effect estimates.

## Conclusion

Exposure to biomass cooking fuel was associated with lower birth weight in our study of term newborns in Kaduna, Nigeria. Data from the nationally representative DHS provide some support for these findings.

## Supporting information

**S1 Table. Sensitivity analysis: 2018 Nigeria DHS maternal and infant characteristics stratified by participation status.**
(DOCX)

**S2 Table. Sensitivity analysis: 2018 Nigeria DHS maternal and infant characteristics stratified by sources of information about birth weight (written card Vs mother's recall).**
(DOCX)

**S3 Table. Association between covariates and birth weight (in grams) among term births, results of linear regression from CEGROMS.**
(DOCX)

**S4 Table. Associations between biomass fuel and birth weight by levels of other factors in 2018 Nigeria Demographic and Health Survey.**
(DOCX)

**S5 Table. Interaction of biomass and child sex by region.**
(DOCX)

**S1 Data. CEGROMS dataset.**
(CSV)

**S2 Data. Nigeria DHS 2018 dataset.**
(CSV)

**S3 Data. CEGROMS data variable labels.**
(XLSX)

**S4 Data. Nigeria DHS 2018 data variable labels.**
(XLSX)

## Acknowledgments

The authors are grateful to the mothers and infants enrolled in the study for their kindness in participating and sharing their experiences of pregnancy and infant growth. We acknowledge the members of the research team for their enthusiasm and perseverance. Child Welfare Clinic, Barau Dikko Teaching Hospital (Kaduna, Nigeria), and staff are appreciated for their help and support. Special mention is Matron Murna Kaffoi, Matron Mairo Mohammed Aliyu, Matron Tani Daniel, and Aisha Salisu who assisted with immunisation during the pilot study. We wish to recognise and thank nutrition and dietetics students involved in growth monitoring during the CEGROMS pilot: Aisha Sa'ad, Olabisi Tawakalitu, Chinoso Endurance Anyaegbu, Sadiya Danbala Isah, Rukayya Muhammad Bello, Charity Peter, Zainab Shamsuddeen, Enemuo Vivian, Emmanuel Mary Eleojo, and Fatima Salisu Bebeji. We are grateful to staff of ERIC Limited, Kaduna, Nigeria for providing technical and logistic support for the data collection.

## Author Contributions

**Conceptualization:** Musa Abubakar Kana, Min Shi, Stephanie J. London.

**Data curation:** Musa Abubakar Kana, Jennifer Ahmed, Abdullahi Yusuf Ashir.

**Formal analysis:** Musa Abubakar Kana, Min Shi, Stephanie J. London.

**Funding acquisition:** Musa Abubakar Kana, Stephanie J. London.

**Investigation:** Musa Abubakar Kana.

**Methodology:** Musa Abubakar Kana, Min Shi, Karimatu Abdullahi, Matthew Taingson, Amina Mohammed-Durosinlorun, Stephanie J. London.

**Project administration:** Musa Abubakar Kana, Jennifer Ahmed, Abdullahi Yusuf Ashir, Halima Bello-Manga, Musa Shuaibu, Abdulkadir Musa Tabari.

**Resources:** Musa Abubakar Kana, Abdullahi Yusuf Ashir, Stephanie J. London.

**Software:** Musa Abubakar Kana, Min Shi, Abdullahi Yusuf Ashir.

**Supervision:** Musa Abubakar Kana, Jimoh Muhammad Ibrahim, Halima Bello-Manga, Musa Shuaibu, Abdulkadir Musa Tabari, Stephanie J. London.

**Validation:** Musa Abubakar Kana, Min Shi, Karimatu Abdullahi, Matthew Taingson, Amina Mohammed-Durosinlorun.

**Visualization:** Musa Abubakar Kana, Min Shi.

**Writing – original draft:** Musa Abubakar Kana.

**Writing – review & editing:** Musa Abubakar Kana, Min Shi, Jimoh Muhammad Ibrahim, Karimatu Abdullahi, Halima Bello-Manga, Matthew Taingson, Amina Mohammed-Durosinlorun, Musa Shuaibu, Abdulkadir Musa Tabari, Stephanie J. London.

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
