## [Decision Letter · Decision Letter 0]

3 Feb 2022

PGPH-D-21-00999

Biomass fuel use and birth weight among term births in Nigeria

Dear Dr Musa Abubakar Kana

Thank you for submitting your manuscript to PLOS Global Public Health. After careful consideration, we feel that it has merit but does not fully meet PLOS Global Public Health’s publication criteria as it currently stands. Therefore, we invite you to submit a revised version of the manuscript that addresses the points raised during the review process.

1. Please check your objective and check why it is different from other studies that used similar analytic approaches to examine the relations of maternal exposure to biomass fuel and birth weight that have already identified in a study conducted by Jiang et al (2016) published in BMC public health.  2. Justify why data of 2018 was used inted of multiple years . 3. Please clarify some missing elements in how the multi-level analysis with DHS was conducted. For example, although authors reported that they adjusted the clustering effects in MLM,  intra-class correlation coefficients and goodness of fit statistics for model improvement was not reported.4. Please explain if a further analysis to examine the association between biomass fuel and a binary outcome of birthweight (underweight vs. normal) is possible to see if the results are similar. If the results are not similar, it should be reported in the manuscript and would enrich the discussion5. The discussion should be enriched  by explaining more in dept some existing interventions, policies and communication strategies to reduce exposure to air pollution, it would be important and interesting for those working in intervention development. Also, a more solid discussion in the research of this area very relevant for the developing countries. Finally, spelling out a little more of these potential implications for practice, will be excellent for awareness on this topic. 

Please submit your revised manuscript in 10 days. If you will need more time than this to complete your revisions, please reply to this message or contact the journal office at globalpubhealth@plos.org. Please include the following items when submitting your revised manuscript:

We look forward to receiving your revised manuscript.

Kind regards,

Marianella Herrera-Cuenca, MD, PhD

Academic Editor

Journal Requirements:

1. Please amend your detailed Financial Disclosure statement. This is published with the article, therefore should be completed in full sentences and contain the exact wording you wish to be published.

Please state what role the funders took in the study. If the funders had no role in your study, please state: “The funders had no role in study design, data collection and analysis, decision to publish, or preparation of the manuscript.”

2. Please ensure that the funders and grant numbers match between the Financial Disclosure field and the Funding Information tab in your submission form. Note that the funders must be provided in the same order in both places as well.

3. Please update your Competing Interests statement. If you have no competing interests to declare, please state: “The authors have declared that no competing interests exist.”

4. Please provide separate figure files in .tif or .eps format only and ensure that all files are under our size limit of 20MB.

Additional Editor Comments (if provided):

Reviewers' comments:

Reviewer's Responses to Questions

**Comments to the Author**

1. Does this manuscript meet PLOS Global Public Health’s publication criteria? Is the manuscript technically sound, and do the data support the conclusions? The manuscript must describe methodologically and ethically rigorous research with conclusions that are appropriately drawn based on the data presented.

Reviewer #1: Yes

Reviewer #2: Yes

2. Has the statistical analysis been performed appropriately and rigorously?

Reviewer #1: No

Reviewer #2: Yes

3. Have the authors made all data underlying the findings in their manuscript fully available (please refer to the Data Availability Statement at the start of the manuscript PDF file)?

Reviewer #1: Yes

Reviewer #2: Yes

4. Is the manuscript presented in an intelligible fashion and written in standard English?

Reviewer #1: Yes

Reviewer #2: Yes

5. Review Comments to the Author

Reviewer #1: This manuscript highlights some important considerations associated with maternal exposure to biomass fuel on reduced infants’ birth weight in northwestern Nigerian population. There are certain components of this manuscript that are strong. The basis established in the background, for instance, is well done. Using nationally representative data in a secondary analysis from the 2018 Nigerian Demographic and Health Survey (DHS), the research team, was able to replicate findings from the study with data of CEGROMS. The paper is easy to read and to-the-point, and I think there is a value in examining such associations in low- and middle-income countries. However, I had several reservations regarding the framing of the research and the methodology. There are some substantive issues that could benefit from clarification or elaboration.

1. In the current study, the authors had the similar research aims, used the similar analytic approaches to examine the relations of maternal exposure to biomass fuel and birth weight that have already identified in a study conducted by Jiang et al (2016) published in BMC public health. So, it is not easy to see what gap is being addressed in the current study except for using the different samples (China vs Nigeria). The authors would need to do a better job explaining what this study adds to that was unclear; what is new here; what are the limitations of the earlier one; how this study different than other studies that preceded this study (e.g., in terms of methodology and conceptual framework).

2. Most substantively, it is not clear to me why only the 2018 DHS was used for replication. The replicate analysis could be extended for other years (e.g., 2013) where the variables of interest are collected in prior DHS. Using data spanning across multiple years may help authors and audience better understand if or how the association between exposure to biomass fuel and birthweight have changed or not changed over time, determine if there are any significant patterns in the association. This approach may help establish the robustness of the current study.

3. Some elements are missing in how the multi-level analysis with DHS was conducted. For example, although authors reported that they adjusted the clustering effects in MLM they did not report intra-class correlation coefficients and goodness of fit statistics for model improvement.

4. Although it seems that the authors mad a good, reasonable decision to treat birth weight as a continuous outcome, I would suggest an additional analysis to examine the association between biomass fuel and a binary outcome of birthweight (underweight vs. normal) to see the results are similar. If the results are not similar, it should be reported in the manuscript.

5. The authors briefly explained some existing interventions, policies and communication strategies to reduce exposure to air pollution. I think that it would be important and interesting for those working in intervention development. Relatedly, they also briefly discuss the potential of implementation science research in this area. A little more discussion here would be helpful about how this might be achieved. In general, the Discussion could be improved by spelling out a little more of these potential implications for practice, possibly including summarising some of the key relevant literature in details.

Reviewer #2: Good article clearly and correctly presented.

This is a very important topic, relevant for the developing world, sometimes neglected by developed nations.

Great approach to understand the impact of the use of biomass fuel in the newborn birth weight.

Statistical analysis is pertinent

Tables and Figures are clear

Conclusions are well supported according to results.

6. PLOS authors have the option to publish the peer review history of their article (what does this mean?). If published, this will include your full peer review and any attached files.

**Do you want your identity to be public for this peer review?** For information about this choice, including consent withdrawal, please see our Privacy Policy.

Reviewer #1: No

Reviewer #2: No

---

## [Editor Report · Decision Letter 1]

22 Apr 2022

Biomass fuel use and birth weight among term births in Nigeria

PGPH-D-21-00999R1

Dear Dr. Musa Abubakar Kana

We are pleased to inform you that your manuscript 'Biomass fuel use and birth weight among term births in Nigeria' has been provisionally accepted for publication in PLOS Global Public Health.

Best regards,

Marianella Herrera-Cuenca, MD, PhD

Academic Editor

Please while in the final phase for publication check mispellings such as plural, the word inmunization is mispelled, change singular to plural such as "stregths and limitations"